# High performance planar germanium-on-silicon single-photon avalanche diode detectors

Peter Vines[1], Kateryna Kuzmenko[1], Jarosław Kirdoda[2], Derek C.S. Dumas[2], Muhammad M. Mirza [2], Ross W. Millar[2], Douglas J. Paul [2] & Gerald S. Buller[1]

Single-photon detection has emerged as a method of choice for ultra-sensitive measurements of picosecond optical transients. In the short-wave infrared, semiconductor-based single-photon detectors typically exhibit relatively poor performance compared with all-silicon devices operating at shorter wavelengths. Here we show a new generation of planar germanium-on-silicon (Ge-on-Si) single-photon avalanche diode (SPAD) detectors for short-wave infrared operation. This planar geometry has enabled a significant step-change in performance, demonstrating single-photon detection efficiency of 38% at 125 K at a wavelength of 1310 nm, and a fifty-fold improvement in noise equivalent power compared with optimised mesa geometry SPADs. In comparison with InGaAs/InP devices, Ge-on-Si SPADs exhibit considerably reduced afterpulsing effects. These results, utilising the inexpensive Ge-on-Si platform, provide a route towards large arrays of efficient, high data rate Ge-on-Si SPADs for use in eye-safe automotive LIDAR and future quantum technology applications.

[1] Institute of Photonics and Quantum Sciences, School of Engineering and Physical Sciences, Heriot-Watt University, Edinburgh EH14 4AS, UK. [2] School of Engineering, University of Glasgow, Rankine Building, Oakfield Avenue, Glasgow G12 8LT, UK. Correspondence and requests for materials should be addressed to P.V. (email: p.vines@hw.ac.uk) or to G.S.B. (email: g.s.buller@hw.ac.uk)

Near room temperature semiconductor-based single-photon avalanche diode (SPAD) detectors have become the accepted optical detection approach in a variety of emerging application areas in the visible and short-wave infrared spectral regions[1–3]. SPAD detectors are avalanche photodiodes biased at fields above avalanche breakdown, in Geiger mode, where a self-sustaining avalanche current can be triggered by an incident single-photon. After the photo-induced avalanche, the detector must be reset, or quenched, before the next detection event. Such detectors typically have a temporal jitter of hundreds of picoseconds, allowing ultra-sensitive measurement of fast optical transients. At wavelengths below 1 µm, silicon-based SPADs have been used in a range of quantum photonic applications, including experiments in quantum foundations[4] and fibre and free-space quantum communications demonstrations[5,6]. In applications such as Light Detection And Ranging (LIDAR), Si-based SPAD detectors have also emerged as a candidate technology due to the high sensitivity and the picosecond temporal response which has resulted in enhanced range and improved surface-to-surface resolution[7]. Si SPAD detectors have been integrated with standard Si CMOS processes to produce ultra-sensitive, large format detector arrays with integrated electronics[8]. This low-cost technology has allowed time of flight systems to be adapted and developed for use in the automotive[9–11] and smartphone industries[12].

There are a number of clear advantages in extending the spectral range of SPAD detectors into the short-wave infrared (SWIR) region, beyond the detection spectrum of Si-based SPADs. Firstly, compatibility with the optical fibre low-loss telecommunications windows is a fundamental advantage in many fibre-based applications. Secondly, in free-space applications such as LIDAR and range-finding, the optical power of the laser source is limited by laser eye-safety thresholds. This eye-safety threshold increases by approximately a factor of 20 or more when the laser wavelength is increased from 850 to 1550 nm (IEC-60825-1 standard), permitting increased optical power whilst maintaining eye-safety in active imaging applications. Consequently, this results in an increased maximum attainable range and/or potential improvements to depth resolution. Thirdly, solar radiation, which typically acts as the background level in most single-photon LIDAR systems, decreases considerably in the SWIR[13]. Finally, operation in the SWIR will mean enhanced atmospheric transmission, .especially through obscurants such as smoke, smog, fog and haze[14,15].

In the SWIR region, the most widely used single-photon detectors are InGaAs/InP SPADs and superconducting nanowire single-photon detectors (SNSPD)[1,2]. Generally, SNSPD devices have had superior single-photon detection performance; however, the cryogenic operating temperatures, typically below 3 K, limit their use in certain key application areas. InGaAs/InP SPADs are the dominant single-photon detector in the SWIR region and have been used in a range of quantum communications experiments, notably in long distance quantum key distribution demonstrations[16]. They are typically operated at temperatures between 220 and 255 K which are achievable using Peltier cooling, which have allowed compact detector modules to be used in LIDAR field trial scenarios[17]. For example, InGaAs/InP SPADs have been used in LIDAR and depth profiling experiments to good effect in the kilometre range[17,18]. Arrays of InGaAs/InP SPADs can give high-performance detection at telecommunications wavelengths[19]; however, two-dimensional arrays may prove expensive for the low-cost, high volume automotive and autonomous vehicle LIDAR markets. One issue that has made use of InGaAs/InP SPADs challenging has been the effect of afterpulsing —described below—which has severely limited the count rates possible of these detectors.

An alternative absorber is the semiconductor material Ge which is sensitive to wavelengths of up to 1600 nm at room temperature, and has the potential to be integrated with Si CMOS[20,21] for integrated electronics with high yield and low cost at volume. SPADs fabricated entirely from Ge have been used for laboratory-based single-photon applications, such as time-resolved photoluminescence[22] but the narrow direct bandgap of Ge of ~0.8 eV meant the sensitivity was limited by high dark count rates (DCR) caused by band-to-band tunnelling. The first Ge-containing SPAD with an Si avalanche region was reported by Loudon et al.,[23] using $Si_{0.7}Ge_{0.3}$/Si multiple quantum wells to absorb the 1210 nm radiation. The efficiency and wavelength range of that detector was limited by the use of a relatively thin, strained superlattice absorbing layer. Although this detector had a poor efficiency in the short-wave infrared, it was an early example of a separate absorption, charge and multiplication (SACM) structure for Si-based SPAD detectors, and conceptually similar to previous SACM designs used in InGaAs/InP SPADs and avalanche photodiodes. Over the last 10 years, it has been possible to grow much thicker epitaxial Ge layers on Si substrates. Kang et al. presented a monolithic Ge-on-Si avalanche photodiode (APD) operating with a high gain bandwidth product[24]. CMOS-compatible Ge waveguide photodiodes[25] and APDs[26] have also been developed. Demonstrations of Ge-on-Si SPADs were presented by Lu et al.[27] and Warburton et al.,[28] the latter reporting a noise equivalent power (NEP) of $1 \times 10^{-14}$ W/Hz$^{1/2}$ at 100 K using a detection wavelength of 1310 nm. More recently, a waveguide Ge-on-Si SPAD was reported by Martinez et al.,[29] with a single-photon detection efficiency (SPDE) of 5.27% at a wavelength of 1310 nm and 80 K. Although promising, none of these SPADs have demonstrated performance that is sufficiently close to the InGaAs/InP SPAD alternative, having high DCR that prohibit their effective use in many applications. To date, normal-incidence Ge-on-Si SPADs have used mesa designs where the sidewalls of the device were exposed. This has placed severe limitations on detector performance due to surface effects resulting in prohibitively high DCRs. These limitations can be avoided by using planar designs where the high field regions are located well away from any sidewalls, mitigating the requirement for high-quality passivation layers. However, planar Ge-on-Si designs are more difficult to fabricate than typical planar InGaAs/InP or Si SPADs due to the Ge/Si lattice mismatch preventing growth of the Si multiplication layer onto the Ge absorber at the top surface of the device.

In this paper we report on the first normal-incidence, planar Ge-on-Si SPAD. We demonstrate high-performance single-photon operation, illustrating high-efficiency detection and low afterpulsing effects. These results illustrate clear potential for integration with Si CMOS for low-cost SPAD array imaging in the SWIR bands.

## Results

**Design, fabrication and preliminary characterisation.** The devices described in this paper use the SACM structure as shown in Fig. 1. The incident SWIR radiation is absorbed in the Ge absorption region and the signal amplification takes place in the Si multiplication region. In between these regions, a selectively implanted charge sheet is used to control the electric field so that the field is high enough in the multiplication region to ensure that avalanche breakdown is reached and low enough in the absorption region to prevent band-to-band and trap-assisted tunnelling. A modest electric field, however, is maintained in the Ge layer to allow efficient drift of photogenerated electrons into the multiplication region. The electric field profile at 5% excess bias above the breakdown voltage is shown in Fig. 2a. The SPAD is biased

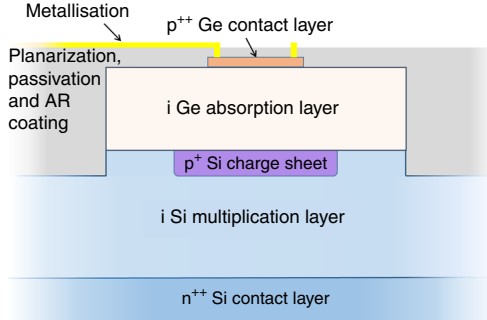

**Fig. 1** Ge-on-Si single-photon avalanche diode (SPAD) structure. Cross-section of a planar Ge-on-Si SPAD showing the Ge top contact and absorption layers, the Si charge sheet, multiplication and bottom contact layers, metallisation and the planarization, passivation and anti-reflection (AR) coating

above the avalanche breakdown voltage and when an incident photon is absorbed in the Ge layer, an electron−hole pair is created and the electron drifts into the Si multiplication region. Here it is accelerated, gaining sufficient kinetic energy to undergo impact ionisation, creating an electron−hole pair. The secondary electrons and holes are in turn accelerated and impact ionise, creating further electron−hole pairs. Further impact ionisation of both holes and electrons rapidly creates a large avalanche current which can be self-sustaining if the device is biased above avalanche breakdown. Under these conditions, this results in a detectable electronic signal that can be timed relative to the initial laser pulse. After detection, it is necessary to bias the SPAD momentarily below avalanche breakdown to quench the avalanche, after which the SPAD can return to its quiescent state ready to detect further incident photons.

Finite element analysis modelling using Silvaco ATLAS software was used to design the SPADs, as shown in Fig. 2a. The charge sheet doping levels and the thicknesses of the multiplication and absorber regions were determined, as well as the optimum overall design dimensions of the SPAD. This was necessary to ensure that the electric field profile throughout the SPAD was appropriate to give high performance, as discussed above. In the simulations shown in Fig. 2a, the electric field profiles are shown for a potential held at 5% above the avalanche breakdown voltage, or 5% excess bias. It is clear from Fig. 2a that there is a low electric field in the Ge absorber at breakdown, and, crucially the high electric field is confined to the centre of the SPAD preventing carriers originating at the sidewalls from causing breakdown events.

The planar SPAD growth and fabrication process is described in the Methods section. A mesa design with exposed sidewalls, similar to Fig. 2b, was also fabricated and used as a control during characterisation. To ensure high yield fabrication, device diameters ranged from 100 to 200 μm, resulting in very large cross-sectional areas compared to previous Ge-on-Si SPADs[27,28]. Future SPADs will be significantly smaller than this—with an aim to reduce the diameter to around 10 μm in order to reduce DCRs further.

Figure 2c shows the dark current at an operating temperature, $T = 100$ K for both the planar and mesa etched SPAD structures fabricated from the same wafer. The mesa etched structure has a similar microstructure to the planar geometry in Fig. 1, except that an etch process was used to create a mesa of diameter less than the diameter of the ion-implanted charge sheet, etched to a depth just below the charge sheet and into the multiplication layer (Fig. 2b). The planar SPAD has a sharp breakdown indicating a low multiplied dark current, previously found to be a strong indicator

of the desired low DCR performance[30]. The mesa etched SPAD has a much softer breakdown with a dark current 50 times higher than the planar structure immediately before breakdown. This indicates that, as expected, significant surface generation is present and suggests that it will have high DCRs compared to the planar SPAD. Indeed, it was not possible to characterise the mesa etched SPAD above breakdown due to its prohibitively high DCR. Figure 2d demonstrates the dark current and photocurrent of the planar SPAD as a function of reverse bias at $T = 78$ K. The dark current before breakdown is less than 1 nA and the SPADs exhibited good uniformity, with little variation in dark current between devices. Photocurrent measurements at a wavelength, $\lambda$, of 1310 nm demonstrate clear punchthrough at 20 V where the electric field reaches the absorption region and photoexcited electrons can drift into the multiplication region. The device yield was over 90%, which at this early stage, is very encouraging for the eventual realisation of Ge-on-Si SPAD focal plane arrays.

**Time-correlated single-photon counting characterisation**. After preliminary characterisation, SPDE, DCR and jitter measurements were taken using the time-correlated single-photon counting (TCSPC) technique, as described in more detail in the Methods section. In these measurements, an electrical gating approach was used to switch the detector to above avalanche breakdown, into the Geiger mode, for a duration of 50 ns in synchronisation with the arrival of the attenuated laser pulse. This gated detector approach was used at a low frequency of 1 kHz in order to fully quench the avalanche and avoid the effects of afterpulsing (described below). The SPAD detectors were initially cooled to $T = 78$ K for SPDE, DCR and jitter measurements using $\lambda = 1310$ nm laser radiation. The SPDE and DCR as a function of excess bias at $T = 78$ K, 100 K and 125 K are shown in Fig. 3. It should be noted that the detectors used had a large area (100 μm diameter), and it is fully expected that the DCR will be considerably lower with reduced area devices, as previously reported in all-Si SPADs[31].

The measured DCR demonstrates a vast improvement when compared to previous Ge-on-Si work. Warburton et al. reported on mesa geometry Ge-on-Si SPADs with a DCR of 5.5 MHz for a 25 μm diameter SPAD at $T = 100$ K[28]. This corresponds to 11,200 counts/s/μm² which is approximately three orders of magnitude higher than the 18.3 counts/s/μm² reported in this work. It should also be noted that, under these conditions, the SPDE reported in that paper is 4%, compared to 26% for the SPAD reported in this paper. There is a similar relationship when our results are compared to results from Martinez et al., who reported a DCR of 500 kHz for a 1 μm wide by 15.9 μm long rectangular waveguide SPAD at $T = 80$ K[29]. This corresponds to 31,400 counts/s/μm² which is over three orders of magnitude higher than the 6.37 counts/s/μm² reported in this work. They report an SPDE of 5%, compared to 22% for the SPAD reported in this paper. This considerable reduction in DCR has resulted from the carefully designed electric field profile of these planar geometry detectors which means that the high electric field is confined within the SPAD, preventing surface states contributing significantly to the DCR. Most dark counts in these SPADs are now likely to originate from dislocations arising from the Si/Ge interface and from thermal excitation throughout the volume of device. In order to fully ascertain the relative contributions to DCR, we are initiating a series of measurements on samples with different diameters and Ge thicknesses.

Figure 3 demonstrates that the SPDE increases with excess bias to a maximum of 38% at $T = 125$ K, significantly higher than previous SPDEs reported for Ge and Ge-on-Si SPADs[22,27−29] and comparable to the highest values recorded for InGaAs/InP SPADs

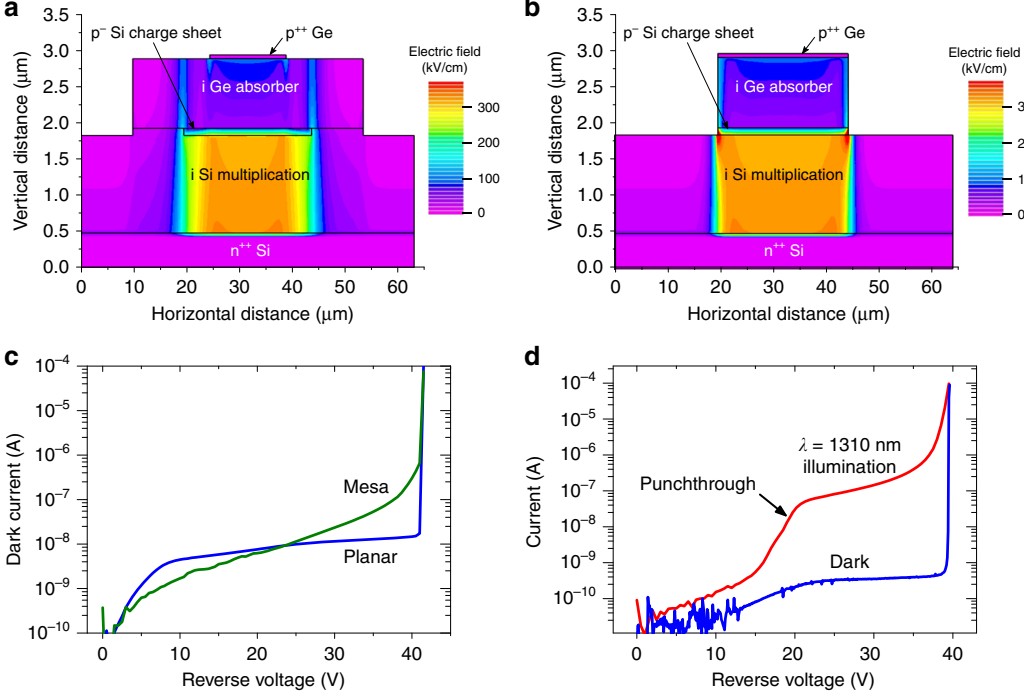

**Fig. 2** Single-photon avalanche diode modelling and current-voltage behaviour. **a** Electric field profile at 5% excess bias for the planar single-photon avalanche diode (SPAD). **b** Electric field profile at 5% excess bias for the mesa etched SPAD. **c** Current-voltage measurements on 100 μm diameter planar (blue line) and mesa etched (green line) structures at a temperature of 100 K under dark conditions. **d** Current-voltage measurements on a 100 μm diameter planar device at a temperature of 78 K. The graph shows dark current (blue line) and photocurrent when illuminated with a 1310 nm laser (red line). Punchthrough is evident at 20 V

at $T = 225$ K[32–34]. This is due in part to the high excess bias applied across the SPAD, attainable due to the low DCR, increasing the breakdown probability in the multiplication region. The relatively thick 1.5 μm Si multiplication region increases the breakdown probability, the likelihood of a self-sustaining avalanche occurring on arrival of the primary electron. The uniform electric field in the multiplication region, caused by minimal residual doping in the lower part of the Si multiplication layer, results in a uniform impact ionisation rate throughout, increasing the breakdown probability still further. The optimised electric field also ensures the efficient transit of photoexcited electrons into the multiplication region. Significantly, there is no conduction band energy barrier between the Ge absorption region and the Si charge sheet ensuring the photoexcited electrons can easily pass between the two regions. Indeed the Si Δ-valley of the conduction band edge is 235 meV below the Ge L-valley conduction band edge in the absorber if calculated using the deformation potentials in ref. [35]. This is an advantage over InGaAs/InP SPADs, which possess an energy barrier step that photoexcited carriers must overcome to reach the InP multiplication region. These SPADs require an InGaAsP grading layer between the InGaAs and InP regions to reduce carrier accumulation at the absorber–charge sheet interface. Finally, an antireflection coating is used to reduce reflection from the top surface of the SPAD to less than 1%. For Ge-on-Si SPADs, the absence of a conduction band barrier at the Ge/Si heterointerface for photogenerated electrons to overcome should ensure that the SPDE remains high at elevated temperatures as the DCR is improved in future design iterations. With these samples, measurements at higher temperatures were limited by the increasing DCR rate due to increasing thermal generation rates; however, a reduction of the detector area in future work is likely to reduce the DCR further and allow a significantly higher operating temperature.

The high SPDE has been achieved despite the use of a relatively thin Ge absorption region. Using absorption coefficients for single crystal Ge at $T = 77$ K[36] we have calculated that less than 50% of the $\lambda = 1310$ nm radiation is absorbed in the 1-μm-thick Ge absorber throughout the operational temperature range. Beer-Lambert's law indicates that a 2-μm-thick Ge absorber will increase the absorption to over 70%, which should lead to an SPDE of greater than 55% at $T = 125$ K. This figure is significantly higher than reported SPDEs for InGaAs/InP SPADs[32–34]. Even thicker Ge layers should provide higher absorption still and we will examine thicker Ge absorbers in future work.

Noise equivalent power (NEP) is a figure of merit for SPADs calculated from the SPDE and DCR of the detector using

$$\text{NEP} = \frac{h\nu}{\text{SPDE}} \sqrt{2\text{DCR}}, \tag{1}$$

where $h$ is Planck's constant and $\nu$ is the frequency of the incident radiation. This can be used to compare detectors, with lower values indicating improved performance. At $T = 78$ K, we have calculated a record low NEP for a Ge-on-Si SPAD detector of $1.9 \times 10^{-16}$ W/Hz$^{1/2}$, 50-fold lower than the NEP of the Ge-on-Si SPAD reported in ref. [28]. NEP values of $3 \times 10^{-16}$ and $7 \times 10^{-16}$ W/Hz$^{1/2}$ were calculated for $T = 100$ K and 125 K respectively.

Figure 4 shows a timing histogram taken at an excess bias of 5.5% and $T = 78$ K when measured at $\lambda = 1310$ nm. The jitter full-width-at-half maximum (FWHM) is 310 ps, which is a reasonable value considering the increased width of the multiplication region[37,38]. Wider multiplication regions generally improve the SPDE but the increased variance in the avalanche build-up time increases the jitter. It is expected that the jitter will reduce as the device diameter is decreased, as found previously in Si SPADs[39], and by improving the electronic packaging of the

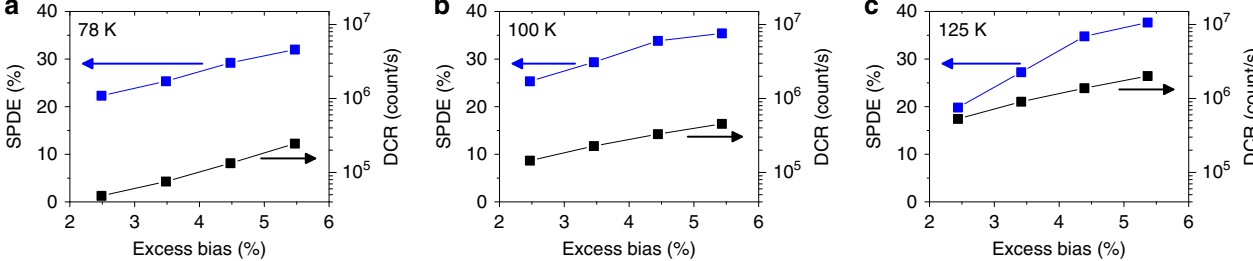

**Fig. 3** Ge-on-Si single-photon avalanche diode (SPAD) performance. Single-photon detection efficiency (SPDE) and dark count rate (DCR) as a function of excess bias for a 100 μm diameter SPAD at temperatures of **a** 78 K, **b** 100 K and **c** 125 K. The measurements were taken using the time-correlated single-photon counting (TCSPC) technique with a 50 ns electrical gate applied to the detector to bias it above avalanche breakdown. For the SPDE measurements the gate was synchronised with the arrival of the 1310 nm wavelength attenuated laser pulse. The detector was gated at a repetition rate of 1 kHz

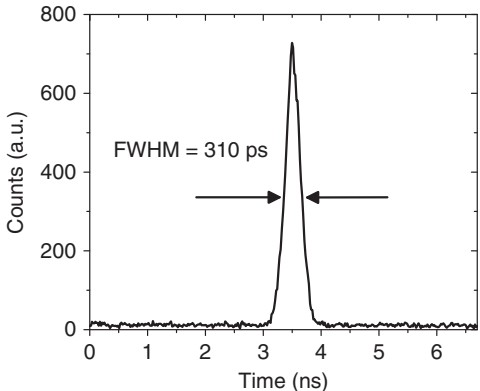

**Fig. 4** Ge-on-Si single-photon avalanche diode (SPAD) jitter. Timing histogram for a 100 μm diameter SPAD at an excess bias of 5.5% and a temperature of 78 K with incident radiation at $\lambda = 1310$ nm. The jitter full-width-at-half maximum (FWHM) was 310 ps

cooled device. Indeed preliminary measurements on Ge-on-Si SPADs with a diameter of 26 μm show a jitter of ~175 ps, closer to the performance of commercial InGaAs/InP SPADs.

The wavelength dependence of the Ge-on-Si SPAD detector efficiency will vary as the Ge bandgap changes with temperature. Figure 5a demonstrates the normalised wavelength dependence of the SPDE as a function of temperature. The high-efficiency SPDE region is related to direct bandgap absorption between the conduction band and the valence bands at the Γ-point. Absorption at longer wavelengths is related to significantly weaker indirect absorption into the L-valleys. At room temperature the direct bandgap of Ge is 0.80 eV[35] but this increases to 0.88 eV at $T = 78$ K, reducing the detection cut-off wavelength[35]. Using a tunable laser we were able to vary the wavelength of the radiation incident on the SPAD from 1310 to 1550 nm to obtain accurate cut-off wavelengths at various temperatures. By defining the cut-off wavelength, $\lambda_c$, as the wavelength at which the detector's SPDE is 50% of the $\lambda = 1450$ nm value. It can be observed that $\lambda_c$ increases from 1468 nm at $T = 125$ K to 1495 nm at $T = 175$ K, increasing at a rate of approximately 0.54 nm/K, in agreement with values calculated using the Varshni temperature-dependent bandgap parameters in ref. [40], and shown in Fig. 5b. Using this device geometry, we expect $\lambda_c$ to reach 1550 nm at $T = 245$ K. However, if the Ge absorber was increased in thickness to 2 μm, $\lambda_c$ will be increased to longer wavelengths and will reach 1550 nm at the lower temperature of 220 K, as shown in Fig. 5c. This should be readily achievable using smaller diameter devices which exhibit lower DCR rates and allow higher temperature operation (as discussed above). This relatively high operating

temperature is achievable using Peltier cooling which will permit Ge-on-Si SPADs to be used effectively in compact, low power LIDAR systems.

**Afterpulsing characterisation.** One critical difference between Ge-on-Si SPADs and the InGaAs/InP SPAD alternative is a realistic potential of a significant reduction in the deleterious effects of detector afterpulsing. This phenomenon occurs when carriers are trapped after an avalanche event and then released later, resulting in an increased background level. Afterpulsing can be mitigated by using a long hold-off time (typically >10 μs) after each event in order that trapped carriers can be released prior to the detector being re-activated. This approach, however, increases the dead-time and restricts the maximum count rate possible. Afterpulsing is recognised as one of the main limitations of InGaAs/InP SPADs, severely affecting their performance, even at modest count rates. Afterpulsing in InGaAs/InP detectors originates mainly in the InP multiplication layer from deep level trap states[41–43], and the expectation with Ge-on-Si SPAD detectors is that the high-quality Si multiplication layer will have a lower density of such states. For the first time, we show a comparison of an InGaAs/InP SPAD with a Ge-on-Si SPAD under nominally identical operating conditions.

Afterpulsing measurements have been performed on a 100 μm diameter Ge-on-Si SPAD at temperatures between 78 and 175 K using the time-correlated carrier counting method[44], where the SPAD undergoes an intentional avalanche. The device is then immediately quenched and then activated via an electrical gate for a second time shortly afterwards. By varying the time between the two detector gates, we examined the probability of an avalanche in the second gate, thus giving the afterpulsing probability as a function of time after the initial avalanche. The results obtained have been compared to a commercial state-of-the-art InGaAs/InP SPAD operating at identical temperatures and applying specific excess biases to each detector in order to obtain an identical SPDE for both detectors. Figure 6a shows the variation in afterpulsing probability of the two SPADs at $T = 125$ K when applying excess biases corresponding to an SPDE of 17% in both detectors. It can be observed that for a specific hold-off time, the afterpulsing probability is significantly reduced for the Ge-on-Si SPAD. For instance, using a hold-off time of 10 μs, the Ge-on-Si SPAD exhibits 20% of the afterpulsing probability of the InGaAs/InP SPAD detector. A similar trend is demonstrated at $T = 150$ K. It should be noted that although the absolute afterpulsing probabilities will be affected by the operating conditions, for example the gate duration, these results serve to provide a comparison between the two detector types under nominally identical operating conditions. These initial results demonstrate considerable promise for further afterpulsing improvement as the Si epilayer material

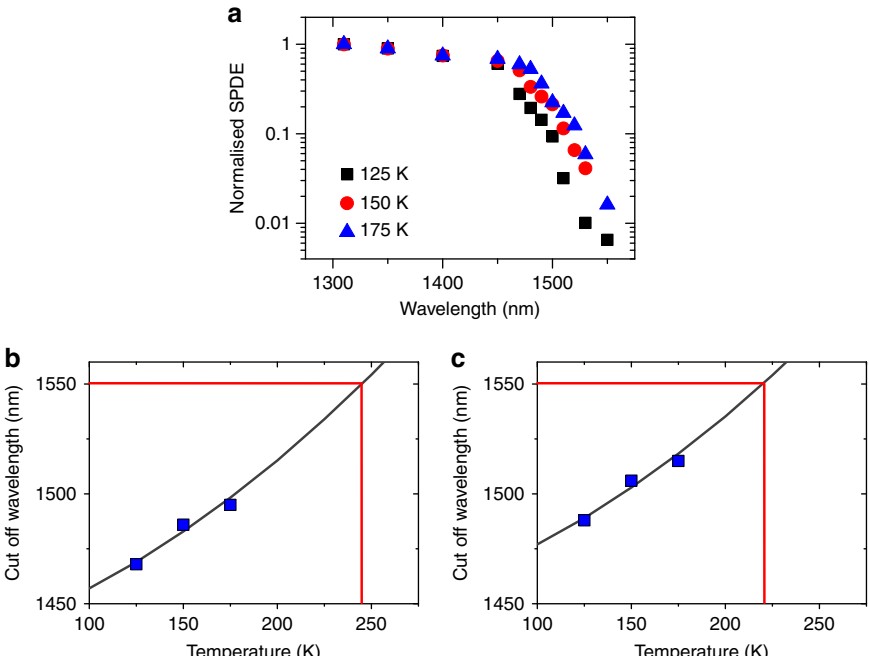

**Fig. 5** Single-photon detection efficiency (SPDE) as a function of wavelength. **a** Normalised SPDE as a function of incident wavelength for a 100 μm single-photon avalanche diode (SPAD) at temperatures of 125 K (black squares), 150 K (red circles) and 175 K (blue triangles). **b** Experimental (squares) and theoretical (line) Ge cut-off wavelength as a function of temperature. **c** Estimated cut-off wavelength and fitting for a Ge-on-Si SPAD with a 2-μm-thick Ge absorption region. The red lines in **b** and **c** indicate the temperature required to reach a cut-off wavelength of 1550 nm

quality, in particular, is improved. Advances in Ge-on-Si SPADs to reduce the DCR will allow further afterpulsing comparisons to be conducted at the higher operating temperatures (>200 K) where InGaAs/InP SPADs are routinely used. At low count rates, the DCR for InGaAs/InP SPADs remains significantly lower than that for Ge-on-Si SPADs. At $T = 125$ K, 1 kHz repetition rate and an SPDE of 17% the 25 μm diameter commercial InGaAs/InP SPAD had a DCR of 3400 counts/s which resulted in an NEP of $7 \times 10^{-17}$ W/Hz$^{1/2}$. In comparison the 100 μm diameter Ge-on-Si SPAD had an NEP of $7 \times 10^{-16}$ W/Hz$^{1/2}$ under the same conditions. Smaller diameter Ge-on-Si SPADs will certainly reduce this difference in sensitivity between SPADs fabricated from these two material systems. In addition, the use of a thicker Ge absorber layer can reduce the DCR further by allowing efficient operation at lower overbias levels.

To investigate the afterpulsing mechanism in Ge-on-Si SPADs, we examined the afterpulse lifetime as a function of temperature in the range 78−125 K, and fitted exponential decays. By fitting Arrhenius plots, we deduced activation energies in the region of 80–90 meV across a range of overbias levels to attempt to ascertain the origin of the traps. Native Si surfaces, native Ge surfaces and GeO$_x$ at Ge surfaces have been shown to have trap states close to 420 meV[45], 130 meV[45] and from 200 to 300 meV[46], respectively. This provides further evidence that the planar geometry is reducing the effects of traps and other impurities at the exposed surfaces. Hence the afterpulsing is unlikely to be related to surface states on the passivated Ge or any exposed Si surfaces.

Figure 6c shows the band structure of the Ge-Si heterointerface calculated at $T = 125$ K using the deformation potentials from ref. [47] without any applied electric field for clarity and includes the three main trapped states for dislocations in Ge plotted as dashed lines inside the bandgap[48]. Whilst the 80−90 meV activation energy extracted from Fig. 6b is close to the 70 meV acceptor and the 90 meV donor trapped states at the valence band edge from the dislocations in the Ge, it is not clear what

mechanism could be responsible for afterpulsing with these states. None of the metal impurities in Ge and Si with energies close to these values are expected to be at any trace levels in the present devices Co (90 meV acceptor), Zn (90 meV acceptor), Hg (87 meV acceptor) and Cr (70 meV acceptor)[49]. Metal impurities in Si with similar energies are Bi (69 meV donor), Ga (72 meV acceptor) and Al (67 meV acceptor)[49]. Whilst Al has been used for the contacts and bond-pads, these are at the top of the Ge and on the back of the Si wafer and the avalanche region of the device is buried so there should be no Al at trace levels close to the avalanche region of the device.

Dislocation traps in Si have been measured to be centred energetically at 807, 870, 940 and 1001 meV above the valence band at $T = 12$ K for the D1−D4 trap states, respectively[50]. The last of these dislocation trap states corresponds to ~130 meV below the Si conduction band at $T = 125$ K[50] (see Fig. 6c) and the linewidth has at least 10 meV of thermal broadening at this temperature. It is well known that during relaxation of a Ge or Si$_{1-x}$Ge$_x$ heterolayer grown above the critical thickness on a Si substrate, some threading dislocations can be injected into the Si substrate[51] and some level of strain will be transferred into the Si close to the heterointerface which will reduce the energy between the trap state and the conduction band edge. The Si D4 trap state has been identified as originating from relaxed dislocations[52] and with the uncertainty of thermal broadening and strain combined with the uncertainty of the afterpulsing excitation energy being extracted from three temperatures, this D4 dislocation trap in Si is a good candidate for the origin of the afterpulsing. Limited area growth of Ge on Si has already demonstrated significant reductions in threading dislocations densities both into the Ge heterolayer and into the Si substrate[53]. This would be one test to determine if the afterpulsing could be further reduced, thereby confirming if the D4 trap state is responsible for the afterpulsing mechanism. Further work is therefore required to confirm this afterpulsing process and to reduce the afterpulsing probability further.

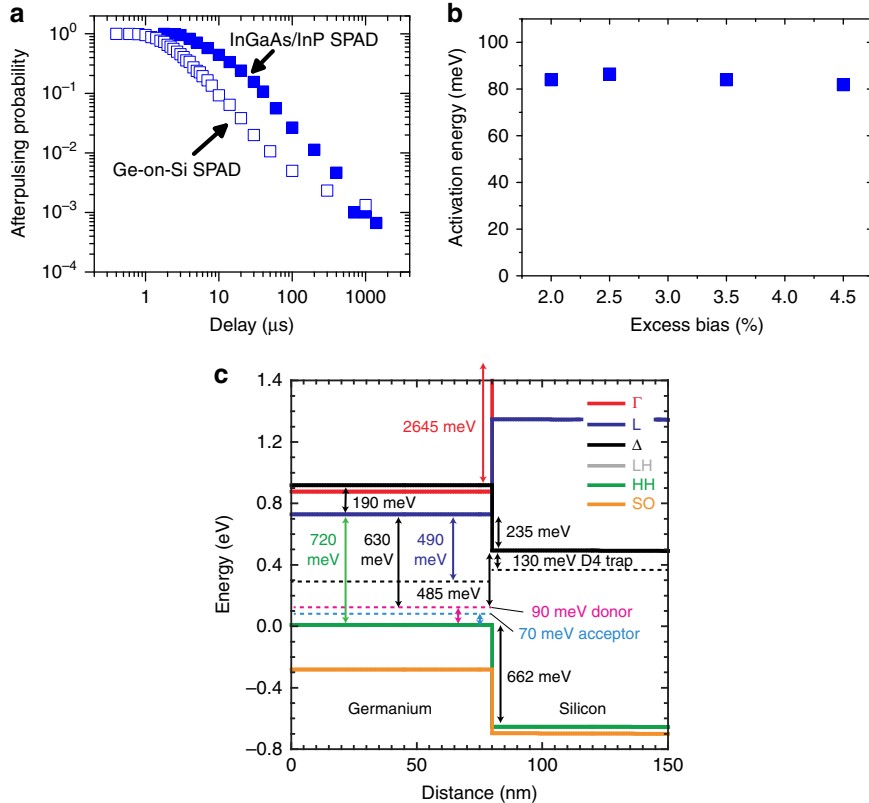

**Fig. 6** Single-photon avalanche diode afterpulsing. **a** Afterpulsing probability as a function of gate delay time for a 100 μm diameter Ge-on-Si single-photon avalanche detector (SPAD) (open squares) compared to a 25 μm diameter commercially available InGaAs/InP SPAD (closed squares) when measured at $\lambda = 1310$ nm and operated at a single-photon detection efficiency (SPDE) of 17% and a temperature of 125 K. **b** Activation energy for the Ge-on-Si SPAD as a function of excess bias. The activation energies were acquired by fitting Arrhenius plots to the afterpulsing lifetime time constants. **c** The band structure at the Ge-Si heterointerface at 125 K showing the trapped defect states from dislocations (dashed lines) and the Γ valley (red line), L valley (blue line), Δ valley (black line), Light Hole (LH) band (grey line), Heavy Hole (HH) band (green line) and Split-Off (SO) band (yellow line)

**Conclusion**. We report high-performance Ge-on-Si SPADs, designed using an innovative planar geometry. SPADs of 100 μm diameter have demonstrated an SPDE of 38% at $T = 125$ K at a detection wavelength of 1310 nm, a significant step-change improvement in the performance levels from all previous reports of Ge-on-Si SPADs, and now comparing more favourably with commercial InGaAs/InP SPADs. The NEP of $1.9 \times 10^{-16}$ W/Hz$^{1/2}$ at $T = 78$ K is a 50-fold improvement on previously reported Ge-on-Si SPADs. Afterpulsing performance has been analysed using the time-correlated carrier counting method and compared to a commercial InGaAs/InP SPAD. With little device optimisation, this initial Ge-on-Si SPAD detector can already operate between a 50 and 75% reduction in dead-time compared to commercial InGaAs/InP SPADs under the same operating conditions, leading to much higher maximum count rates. These results point to a clear route to smaller volume detectors incorporating thicker Ge absorbers being capable of operation at, or near, room temperature, with low DCR, low afterpulsing and high count rate operation. The increased temperature will also allow these detectors to operate with high-efficiency at 1550 nm wavelength. The use of a Si platform provides a low-cost route for single-photon three-dimensional imaging and sensing in the eye-safe short-wave infrared region. This could have significance for a range of commonplace applications such as automotive and autonomous vehicle LIDAR, security and environmental LIDAR monitoring in addition to enabling a range of quantum technology applications that use the important telecommunications wavelengths.

## Methods

**Device fabrication**. Five structures were grown on 150 mm diameter n$^{++}$-doped Si (001) substrates. Firstly, a 1.5-μm-thick Si multiplication region was grown by a commercial reduced pressure chemical vapour deposition (RP-CVD) system. Finite element analysis modelling using Silvaco ATLAS software was used to determine the optimum charge sheet density to provide low electric fields in the Ge absorber and high electric fields in the Si avalanche region. Photolithography was used to define the charge sheet regions which were then implanted with boron acceptors at an energy of 10 keV. Different charge sheet doses were implanted in each of the five wafers to account for fabrication tolerances and ensure that the optimised electric field profile was achieved. After implantation the boron dopants were activated at 950 °C for 30 s using a rapid thermal annealer. After Radio Corporation of America (RCA) cleaning, a 1-μm-thick, nominally undoped Ge absorption region and a 50 nm p$^{++}$ Ge top contact layer were grown on top of the selectively implanted Si layer using RP-CVD. A trench etch through the Ge was performed at a lateral distance of 10 μm from the charge sheet, in order to electrically isolate the SPADs, as shown in Fig. 1. This electrical isolation was required due to the conductive path formed by the background doping level found in the Ge layer. Metal contacts, GeO$_2$ passivation, anti-reflection (AR) coatings and bond-pads were subsequently deposited.

**Single-photon characterisation**. SPDE, DCR and jitter measurements were taken using the TCSPC technique, with a schematic of the setup used for this characterisation shown in Fig. 7. The SPAD detector was mounted in an Oxford Instruments liquid nitrogen cryostat that enabled measurements between $T = 78$ K and $T = 175$ K. The cryostat has optical access, which allowed a short working distance between the detector and the external optical system. Picosecond pulsed lasers were used to measure the SPDE of the SPAD, with the output attenuated to a level of much less than one photon per pulse in order to reduce the probability of a single pulse containing more than one photon. Two lasers were used—a PicoQuant laser diode emitting a wavelength of 1310 nm as well as an NKT Supercontinuum laser tunable in the wavelength range between 1150 and 2000 nm. The laser outputs were coupled into single mode fibres (SMF-28) and then to a 50:50 fibre splitter. One splitter output was connected to a calibrated power meter to provide in situ power readings, which were continuously monitored during the measurements. The other output was connected to a programmable optical attenuator where it could be attenuated by up to 100 dB.

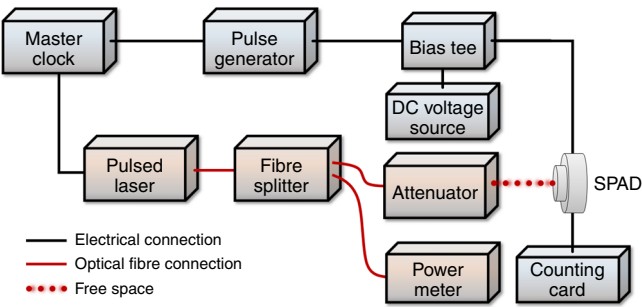

**Fig. 7** Single-photon characterisation setup. Schematic diagram of the experimental setup used for time-correlated single-photon counting (TCSPC) characterisation. The black lines denote electrical connections, the red solid lines denote optical fibre connections and the red dotted line denotes the free-space optical connection. The laser spot was directed onto the optical area of the SPAD using a broadband imaging system (not shown). The cryostat housing the SPAD has been removed for clarity

This attenuator output was inserted into a reflective collimator for free-space collimation. The collimated beam passed through two pellicle beamsplitters (92:8 splitting ratio), which allowed a broadband illumination channel and an imaging channel to an SWIR camera to be used to align and focus the laser spot on the SPAD. Prior to each SPDE measurement, the laser power reaching the SPAD was measured using a calibrated optical power meter and compared to the power measured by the in situ power meter. This ratio was used to calculate the number of photons per pulse incident upon the SPAD for a given power meter reading. The devices were operated in gated mode with a direct current (DC) voltage source biasing the SPAD just below the breakdown voltage. A master clock controlled the timing of the laser trigger, the detector gate and the start signal for the Edinburgh Instruments TCC900 photon counting card. A detector gate of 50 ns duration was used in the experiments described in this paper. The detector gate was combined with the DC bias using a Tektronix 5530 Bias Tee, with the output connected to the anode of the SPAD. The SPAD cathode was connected to the photon counting card stop signal. Timing histograms with a timing bin width of 19.5 ps recorded by the photon counting card were used to calculate the device jitter.

## Data availability

All relevant data are available from the Heriot-Watt University data archive[54].

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

## Acknowledgements

The authors wish to acknowledge the support of UK EPSRC projects EP/N003446/1, EP/K015338/1, EP/L024020/1, EP/M01326X/1, EPN003225/1 and the DSTL Ph.D. scholarship DSTLX-1000092774.

## Author contributions

The planar Ge-on-Si SPAD device was proposed by R.W.M. and D.J.P., with later significant input from G.S.B. and P.V. The design was subsequently developed by all the authors. The devices were fabricated by J.K., D.C.S.D. and M.M.M. using a process developed by J.K., D.C.S.D., R.W.M., M.M.M. and D.J.P. P.V. and K.K. undertook the Silvaco TCAD simulations, and the experimental characterisation and data analysis, under the supervision of G.S.B. All authors discussed the results and approved the final manuscript.

## Additional information

**Competing interests:** The authors declare no competing interests.

