## [Peer Review File · Nature Communications]

Reviewers' comments:

Reviewer #1 (Remarks to the Author):

The authors present in their manuscript "High Performance Ge-on-Si single-photon avalanche Diode detectors" remarkable research results on low-light level detectors and achieved significant progress in the field of fast NIR SPAD development. The manuscript is superbly written and is about a very interesting topic that will definitely attract a wider readership.

Nevertheless, there are a few points to think about.

1st: In the introduction, I would like to see a reference to MIT, the work of J. Michel.

>> Michel, J., J. Liu, and L. C. Kimerling. "High-performance Ge-on-Si photodetectors." *Nature Photonics* 4.8 (2010): 527.

2nd: Figure 5 b+c: Can you highlight the differences in the figures (not only in caption)? e.g. Title: b)"Ge cut-off" / c)"Ge-on-Si cut-off (2 μ m Ge layer)"

Reviewer #2 (Remarks to the Author):

The authors present a complete description of new high-performance single photon detector in the Ge on Si platform. They report excellent performance metrics for a large area planar separate absorber charge multiplication Ge on Si Geiger mode APD's. Indeed, this represents the first to my knowledge Ge on Si SPD with performance competitive or equal to InP/InGaAs APS's state of the art SPAD's. The main advantage of this device is the CMOS compatible nature of the fabrication of the device and the potential to be directly integrated with readout electronics. It is anticipated that by scaling the mesa size below 100 microns will reduce the dark count rate and further increase the device performance. I have one question that I did not see addressed in the text. Is this mainly due to dark counts arising from the perimeter or is it due to the Si/Ge interface?

I find this work to be a significant advance and due to the area of application being Geiger mode LIDAR, I would highly recommend for publication in *Nature Communications*.

Reviewer #3 (Remarks to the Author):

This is an interesting paper in which the authors report on a "planarized" Ge on Si single photon avalanche diode (SPAD). The authors have, as claimed, achieved breakthrough performance relative to previous Ge on Si SPADs. They also compare their device to the other SPAD that is frequently used in the SWIR spectral window, InP/InGaAs separate absorption, charge, and multiplication SPADs. The Ge on Si has the advantage relative to InP/InGaAs of being compatible with Si electronics for circuit arrays. There is a potential cost advantage as well, however, contrary to popular belief, chip cost is seldom a major cost factor. The detection efficiency of 38% is excellent. Compared with InP/InGaAs SPADs the dark count rate, jitter, and afterpulsing are not very impressive. Also, cooling to 100K is a major disadvantage for any practical implementation. Nevertheless, this is a good paper that brings Ge on Si into viable competition. I recommend publication, however, I request that the authors address the following:

1. The authors compare their single photon detection efficiency and afterpulsing to InP/InGaAs, however, they fail to compare their dark count rate.
2. They also fail to note that the InP/InGaAs SPADs typically operate at 245K to 280K, whereas their device is cooled to 100K or 125K. This is a significant issue since the temperatures for the InP/InGaAs can be achieved with a two or three stage thermoelectric cooler, which is not possible for their SPAD. This makes a big difference for eventual deployment in LIDAR. The authors

speculate that they can achieve higher temperatures, however, there is a big difference in their current temperature and their projected temperature. It should also be noted in their projections that detection efficiencies often decrease with increasing temperature.

3. The jitter of the Ge on Si SPADs is higher than state-of-the-art by a significant amount. The authors speculate that the jitter can be improved by decreasing the device diameter. That is presumably, but not stated, due to the fact that avalanches are initiated in one spot and travel laterally across the device. This has been reported and can be easily modeled. It would be useful if the authors provided an approximate calculation of the improvement that can be achieved.

4. The authors state that their device can be used to wavelengths as long as 1550 nm. However, even at room temperature the absorption coefficient of Ge drops rapidly just above 1550 nm. The authors' suggestion that this device will ever be viable at 1550 nm is suspect. I suggest the authors should confine their comments to 1300 nm where they carried out their measurements.

5. The authors' comparison of afterpulsing between Ge on Si and InP/InGaAs is not a fair comparison. As stated above, InP/InGaAs is typically operated at 150K higher temperature than the Ge on Si APDs reported in this paper. Afterpulsing tends to decrease with increasing temperature. Therefore for a fair comparison, the Ge on Si SPAD afterpulsing should be compared with InP/InGaAs afterpulsing at ~ 250K.

Authors' response to the reviewers' comments:

We would like to thank the reviewers for their time and for the useful comments they have made which will undoubtedly help us to improve the manuscript. We have highlighted the changes we have made to the manuscript in yellow.

Reviewer #1 (Remarks to the Author):

The authors present in their manuscript "High Performance Ge-on-Si single-photon avalanche Diode detectors" remarkable research results on low-light Level detectors and achieved significant progress in the field of fast NIR SPAD development. The manuscript is superbly written and is about a very interesting topic that will definitely attract a wider readership.

Nevertheless, there are a few points to think about.

1st: In the introduction, I would like to see a reference to MIT, the work of J. Michel.

>> Michel, J., J. Liu, and L. C. Kimerling. "High-performance Ge-on-Si photodetectors." Nature Photonics 4.8 (2010): 527.

The reviewer makes a good point regarding other work in the area. We have added this reference to page 2, paragraph 3, line 2 of the manuscript: "At room temperature, Ge is sensitive to incident radiation with wavelengths of up to 1600 nm and has the potential to be integrated with Si CMOS ²¹₂₂."

2nd: Figure 5 b+c: Can you high light the differences in the figures (not only in caption)? e.g. Title: b)"Ge cut-off" / c)"Ge-on-Si cut-off (2μm Ge layer)"

We have altered figure 5 as specified.

Reviewer #2 (Remarks to the Author):

The authors present a complete description of new high-performance single photon detector in the Ge on Si platform. They report excellent performance metrics for a large area planar separate absorber charge multiplication Ge on Si Geiger mode APD's. Indeed, this represents the first to my knowledge Ge on Si SPD with performance competitive or equal to InP/InGaAs APS's state of the art SPAD's. The main advantage of this device is the CMOS compatible nature of the fabrication of the device and the potential to be directly integrated with readout electronics. It is anticipated that by scaling the mesa size below 100 microns will reduce the dark count rate and further increase the device performance. I have one question that I did not see addressed in the text. Is this mainly due to dark counts arising from the perimeter or is it due to the Si/Ge interface?

The planar design of the SPADs in this paper has removed the high field region from the edge of the detector greatly reducing the dark counts originating from the perimeter. Therefore most of the remaining dark counts will originate from the thermal events within the Ge absorber and Si

multiplication layer and from the Si/Ge interface. Measurements on a range of SPADs with different detector diameters ranging from 26 to 200 μm are being conducted to investigate this further and these results will be presented in future work.

We have added “Most dark counts in these SPADs are now likely to originate from dislocations arising from the Si/Ge interface and from thermal excitation throughout the volume of device. In order to fully ascertain the relative contributions to dark count rate, we are initiating a series of measurements on samples with different diameters and Ge thicknesses.” to the end of paragraph 2, page 6.

I find this work to be a significant advance and due to the area of application being Geiger mode LIDAR, I would highly recommend for publication in Nature Communications.

Reviewer #3 (Remarks to the Author):

This is an interesting paper in which the authors report on a “planarized” Ge on Si single photon avalanche diode (SPAD). The authors have, as claimed, achieved breakthrough performance relative to previous Ge on Si SPADs. They also compare their device to the other SPAD that is frequently used in the SWIR spectral window, InP/InGaAs separate absorption, charge, and multiplication SPADs. The Ge on Si has the advantage relative to InP/InGaAs of being compatible with Si electronics for circuit arrays. There is a potential cost advantage as well, however, contrary to popular belief, chip cost is seldom a major cost factor. The detection efficiency of 38% is excellent. Compared with InP/InGaAs SPADs the dark count rate, jitter, and afterpulsing are not very impressive. Also, cooling to 100K is a major disadvantage for any practical implementation. Nevertheless, this is a good paper that brings Ge on Si into viable competition. I recommend publication, however, I request that the authors address the following:

1. The authors compare their single photon detection efficiency and afterpulsing to InP/InGaAs, however, they fail to compare their dark count rate.

The reviewer is correct; in order to give a more complete comparison it is also important to compare the dark count rate of the InGaAs/InP and Ge-on-Si SPADs and its effect on sensitivity. We have added “At low count rates, the DCR for InGaAs/InP SPADs remains significantly lower than that for Ge-on-Si SPADs. At 125 K, 1 kHz repetition rate and an SPDE of 17 % the 25 μm diameter commercial InGaAs/InP SPAD had a DCR of 3400 counts/s which resulted in a NEP of $7 \times 10^{-17} \text{ WHz}^{-1/2}$. In comparison the 100 μm diameter Ge-on-Si SPAD had an NEP of $7 \times 10^{-16} \text{ WHz}^{-1/2}$ under the same conditions. Smaller diameter Ge-on-Si SPADs will certainly reduce this difference in sensitivity between SPADs fabricated from these two material systems. In addition, the use of a thicker Ge absorber layer can reduce the DCR further by allowing efficient operation at lower overbias levels.” to the end of paragraph 1, page 10.

2. They also fail to note that the InP/InGaAs SPADs typically operate at 245K to 280K, whereas their device is cooled to 100K or 125K. This is a significant issue since the temperatures for the InP/InGaAs can be achieved with a two or three stage thermoelectric cooler, which is not possible

for their SPAD. This makes a big difference for eventual deployment in LIDAR. The authors speculate that they can achieve higher temperatures, however, there is a big difference in their current temperature and their projected temperature. It should also be noted in their projections that detection efficiencies often decrease with increasing temperature.

The reviewer has correctly described that we have failed to mention the typical temperature range of current generation InGaAs/InP SPADs and the importance of operational temperature to a range of applications, including LIDAR measurements. This was an oversight on our part, and we have corrected our error by inserting “They are typically operated at temperatures between 220 K and 255 K which are achievable using Peltier cooling, which have allowed compact detector modules to be used in LIDAR field trial scenarios¹⁸.” to paragraph 2 on Page 2.

We agree with the reviewer that the Ge-on-Si devices operate at low temperature, and that this is not fully compatible with a practical LIDAR demonstration, and we have made a small addition on Page 8: “This relatively high temperature operation is achievable using Peltier cooling which will permit Ge-on-Si SPADs to be used effectively in compact, low power LIDAR systems.”

In the results presented in fig. 3 of this paper demonstrate that over the measured temperature range the SPDE increases slightly as temperature is increased. Studies on InGaAs/InP SPADs often show a similar trend¹. The relevant figure from reference 1 has been reproduced below. Since there is no conduction band barrier in Ge-on-Si SPADs to prevent the photo-excited electron from initiating the self-sustaining avalanche process we expect the SPDE to remain reasonably constant as higher operating temperatures are achieved.

Fig. 1 – Reproduced from Pellegrini *et al.*¹ showing dependence of SPDE with temperature for a variety of InGaAs/InP SPADs.

We have added “For Ge-on-Si SPADs, the absence of a conduction band barrier at the Ge/Si hetero-interface for photogenerated electrons to overcome should ensure that the SPDE remains high at elevated temperatures as the DCR is improved in future design iterations.” to paragraph 1, page 7 of the manuscript.

3. The jitter of the Ge on Si SPADs is higher than state-of-the-art by a significant amount. The authors speculate that the jitter can be improved by decreasing the device diameter. That is presumably, but not stated, due to the fact that avalanches are initiated in one spot and travel laterally across the device. This is has been reported and can be easily modeled. It would be useful if the authors provided an approximate calculation of the improvement that can be achieved.

The reviewer is correct in that the jitter presented is higher than most reports of InGaAs/InP SPADs, which are generally of smaller diameter and have correspondingly lower jitter as a result of the more rapid lateral evolution of the avalanche. However, since submission, we have made measurements on smaller device diameters with the same microstructure. These recent measurements indicate that with Ge-on-Si SPADs with a diameter of 26 μm a jitter of 175 ps is possible, significantly less than the 310 ps reported in this manuscript. We have added “Indeed preliminary measurements on Ge-on-Si SPADs with a diameter of 26 μm show a jitter of ~ 175 ps, closer to the performance of commercial InGaAs/InP SPADs.” to page 7 of the manuscript.

4. The authors state that their device can be used to wavelengths as long as 1550 nm. However, even at room temperature the absorption coefficient of Ge drops rapidly just above 1550 nm. The authors’ suggestion that this device will ever be viable at 1550 nm is suspect. I suggest the authors should confine their comments to 1300 nm where they carried out their measurements.

As shown in fig. 5 of the manuscript, the detector cut-off wavelength increases with temperature. This is shown both experimentally and using temperature and strain dependent bandgap parameters, values which are in good agreement. All Ge-on-Si heterolayers are tensile strained at $\sim 0.2\%$ due to the different coefficients of expansion with temperature which red-shifts the direct and indirect bandgap absorption to longer wavelengths. The authors are therefore confident the Ge-on-Si SPADs will be sensitive to 1550 nm radiation when operated at the temperatures quoted in the manuscript. Higher operating temperatures can be achieved by designing smaller area detectors, using optimised Ge absorber layers and by using limited area growth as stated on pages 8 and 12 of the manuscript.

5. The authors’ comparison of afterpulsing between Ge on Si and InP/InGaAs is not a fair comparison. As stated above, InP/InGaAs is typically operated at 150K higher temperature than the Ge on Si APDs reported in this paper. Afterpulsing tends to decrease with increasing temperature. Therefore for a fair comparison, the Ge on Si SPAD afterpulsing should be compared with InP/InGaAs afterpulsing at $\sim 250\text{K}$.

The reviewer correctly states that the ideal afterpulsing comparison should be conducted at temperatures where InGaAs/InP SPADs are routinely operated, and that this manuscript concentrates on the operating temperatures of Ge-on-Si SPADs – i.e. below 200 K. Due to their relatively immature stage of development we are currently unable to demonstrate Ge-on-Si with adequate performance at temperatures consistent with Peltier cooling. It is certainly true that the effects of afterpulsing are reduced at increased temperatures as the trap lifetime reduces, however this reduction will occur for both SPAD material types. We maintain that we have made the best

comparison possible by measuring both SPAD types under the identical conditions of efficiency, temperature and gating conditions, a comparison that has never before been possible.

We have added “*Advances in Ge-on-Si SPADs to reduce the DCR will allow further afterpulsing comparisons to be conducted at the higher operating temperatures (>200 K) where InGaAs/InP SPADs are routinely used.*” to page 10 of the manuscript.

References

[1] Pellegrini *et al.*, Design and Performance of an InGaAs–InP Single-Photon Avalanche Diode Detector, *IEEE J. Quantum Electronics* **42**, 397-403 (2006).

REVIEWERS' COMMENTS:

Reviewer #2 (Remarks to the Author):

I feel the authors have addressed my concerns and I feel the manuscript is acceptable in Nature Comm.

Reviewer #3 (Remarks to the Author):

The authors have provided satisfactory responses to the reviews. I think the paper is now acceptable for publication.